# The Cell Polarity Protein MPP5/PALS1 Controls the Subcellular Localization of the Oncogenes *YAP* and *TAZ* in Liver Cancer

**DOI:** 10.3390/ijms26020660

**Published:** 2025-01-14

**Authors:** Marcell Tóth, Shan Wan, Jennifer Schmitt, Patrizia Birner, Teng Wei, Fabian von Bubnoff, Carolina de la Torre, Stefan Thomann, Federico Pinna, Peter Schirmacher, Sofia Maria Elisabeth Weiler, Kai Breuhahn

**Affiliations:** 1Institute of Pathology, Medical Faculty Heidelberg, Heidelberg University, 69120 Heidelberg, Germany; 2Diagnostic and Research Institute of Pathology, Medical University of Graz, 8010 Graz, Austria; 3Department of Pathology, Suzhou Medical College of Soochow University, Suzhou 215123, China; 4Cytotherapy Laboratory, Shenzhen People’s Hospital, Shenzhen 518020, China; 5NGS Core Facility, Medical Faculty Mannheim, Heidelberg University, 68167 Mannheim, Germany; 6Würzburg Institute of Systems Immunology, Julius-Maximilians-Universität Würzburg, 97078 Würzburg, Germany

**Keywords:** tumor-suppressor gene, apical polarity, Crumbs complex, HCC, Hippo pathway

## Abstract

The oncogenes yes-associated protein (*YAP*) and transcriptional coactivator with PDZ-binding motif (*TAZ*) are potent liver oncogenes. Because gene mutations cannot fully explain their nuclear enrichment, we aim to understand which mechanisms cause *YAP/TAZ* activation in liver cancer cells. The combination of proteomics and functional screening identified numerous apical cell polarity complex proteins interacting with YAP and TAZ. Co-immunoprecipitation (Co-IP) experiments confirmed that membrane protein palmitoylated 5 (MPP5; synonym: PALS1) physically interacts with YAP and TAZ. After removing different MPP5 protein domains, Co-IP analyses revealed that the PDZ domain plays a crucial role in YAP binding. The interaction between YAP and MPP5 in the cytoplasm of cancer cells was demonstrated by proximity ligation assays (PLAs). In human hepatocellular carcinoma (HCC) tissues, a reduction in apical MPP5 expression was observed, correlating with the nuclear accumulation of YAP and TAZ. Expression data analysis illustrated that *MPP5* is inversely associated with YAP/TAZ target gene signatures in human HCCs. Low *MPP5* levels define an HCC patient group with a poor clinical outcome. In summary, MPP5 facilitates the nuclear exclusion of YAP and TAZ in liver cancer. This qualifies MPP5 as a potential tumor-suppressor gene and explains how changes in cell polarity can foster tumorigenesis.

## 1. Introduction

Cell polarity, particularly epithelial cell polarity, is essential for the structural and functional integrity of tissues and organs. Polarity enables epithelial cells to establish apical–basal and planar organization, forming boundaries that protect against external agents, mediate selective permeability, and facilitate complex functions like secretion and absorption. This polarity is regulated by conserved polarity complexes, notably the Crumbs, Par, and Scribble complexes, which interact with cell–cell junctions, including adherens and tight junctions, to maintain polarity and tissue architecture [1,2]. Especially for liver hepatocytes, protein secretion and bile production depend on a unique multipolar organization essential for countercurrent blood and bile flow [3,4].

Loss or alteration of cell polarity is one of the earliest events in carcinogenesis, suggesting that polarity proteins play a role beyond spatial cell orientation. Indeed, evidence shows that in early tumor development, the disruption of epithelial architecture coincides with the disorganization of cell polarity complexes [5]. For example, constituents of the Crumbs and Scribble complexes are frequently mislocalized or downregulated in various carcinomas, including hepatocellular carcinoma (HCC) [5]. However, our understanding of the underlying mechanisms that link polarity disruption with the aberrant activation of signaling pathways and tumorigenesis remains fragmented.

Previous data suggest that polarity proteins may influence cancer initiation and progression through their interactions with the Hippo signaling pathway, an evolutionarily conserved regulator of cell proliferation, organ size, and apoptosis [6]. The Hippo pathway consists of a kinase cascade that modulates two downstream effectors, yes-associated protein (YAP) and transcriptional co-activator with PDZ-binding motif (TAZ). These transcriptional co-activators bind transcription factors of the TEA Domain Family Member (TEAD family) and control gene expression. Active Hippo signaling phosphorylates YAP and TAZ in non-malignant cells, retaining them in the cytoplasm and preventing their translocation to the nucleus. In many cancers, disturbed Hippo signaling or YAP/TAZ overexpression results in the nuclear accumulation of both factors and subsequent induction of tumor-supporting genes [7]. Notably, the role of the Hippo pathway in liver cancer has gained substantial attention because elevated nuclear YAP/TAZ levels correlate with poor prognosis in HCC [8]. The clinical relevance of the Hippo/YAP/TAZ signaling cascade is underscored by the development of various YAP/TAZ/TEAD inhibitors, some of which are already undergoing clinical testing [9,10]. The oncogenic potential of *YAP* and *TAZ* in hepatocarcinogenesis has also been demonstrated in various mouse models [11,12,13]. For example, the inducible overexpression of YAP/TAZ in hepatocytes or the genetic inactivation of Hippo pathway components such as mammalian STE20-like protein kinase 1/2 leads to the rapid induction of aggressive liver tumors [11,12,13].

Notably, components of the polarity complexes, such as Scribble and Crumbs, can attenuate YAP/TAZ activity. For instance, Scribble has been shown to bind to YAP, preventing its nuclear localization and inhibiting its pro-proliferative effects. This spatial control by polarity proteins suggests that disrupting polarity could indirectly deactivate Hippo signaling and enhance the nuclear enrichment of YAP/TAZ, creating a permissive environment for tumorigenesis [14]. Despite these findings, a comprehensive understanding of how polarity proteins regulate Hippo signaling, particularly in hepatocyte-derived cancer cells, is still lacking. While some data suggest that polarity protein mislocalization can lead to Hippo pathway dysregulation, the full spectrum of interactions and regulatory mechanisms involved remains unknown. This knowledge gap is critical, as understanding how polarity proteins influence YAP/TAZ activity could provide insights into liver cancer progression and identify new therapeutic targets within the Hippo/YAP/TAZ axis.

Through protein binding studies and functional assays, we identify the Crumbs cell polarity complex as a critical factor in the cytoplasmic retention of YAP and TAZ. We describe the PDZ domain in membrane protein palmitoylated 5 (MPP5) as the site relevant to the interaction between MPP5 and YAP. Inactivation of *MPP5* leads to the translocation of YAP into the nucleus and the induction of target gene expression. We also demonstrated this inverse relationship between MPP5 expression and YAP activation in HCC patient tissues, where low *MPP5* expression is associated with a worse patient prognosis. In summary, our data demonstrate that the Crumbs cell polarity complex, particularly the protein MPP5, plays a crucial role in regulating the subcellular localization of Hippo pathway effectors.

## 2. Results

### 2.1. Identification of YAP/TAZ Upstream Regulators

To identify potential cellular structures that physically bind YAP and/or TAZ in HCC cells, we analyzed our recently published interactome data for both proteins [15]. These data illustrated that both factors interact with proteins involved in, e.g., signaling pathways, cytoskeletal remodeling, and cytokinesis. For these processes, the number of YAP interaction partners was much higher than for TAZ, although BirA-tagged YAP and TAZ were expressed equally. The diversity of the interactome suggests that YAP partially performs functions distinct from those of TAZ. Notably, almost all detected proteins with cell polarity or cell–cell contact functions do not show a preference for binding YAP, as they interact equally with YAP and TAZ (Figure 1). These results indicated that both factors partly facilitate their tumor-supporting functions through the differential binding of, e.g., transcriptional regulators or proteins involved in cellular signaling, migration, or cell division [16]. In contrast, similar mechanisms may control the subcellular localization and activity of YAP and TAZ.

Because we were particularly interested in the regulatory mechanisms that control YAP/TAZ localization, we focused on cell polarity factors. However, the interaction between YAP/TAZ and cell polarity complexes does not prove to be a regulatory mechanism in cancer cells, which are known to lose their spatial organization. For this reason, we performed a functional assay after inhibiting different constituents of the polarity complexes in liver cancer cells (Figure 2A). We utilized HepG2 cells, known for their high differentiation, cell polarity, and the preservation of most hepatocyte-specific metabolic functions [17,18]. These cells express significant amounts of YAP/TAZ and were used for a systematic small interfering RNA (siRNA) screen of 26 proteins, including components of the apical, lateral, and basolateral polarity complexes (the Crumbs, Par, and Scribble complexes) (Appendix A) [1,16]. In addition, other factors indirectly controlling the spatial organization of cells were included in this analysis (e.g., p21 (RAC1) activated kinase 1 (PAK1) with cytoskeleton-modifying properties) (Figure 2A) [19]. Subsequent immunofluorescence analysis of endogenous YAP and TAZ revealed that the silencing of 16/26 polarity proteins caused a significant nuclear enrichment of at least one of both proteins (Figure 2A). Notably, the knockdown of all apical Crumbs complex factors led to a nuclear accumulation of YAP/TAZ, while only one lateral Par-6 family cell polarity regulator gamma (PARD6G) and three basolateral (catenin beta 1 (CTNNB1), discs Large MAGUK scaffold protein 3 (DLG3), LLGL Scribble cell polarity complex component 2 (LGL2)) complex constituents caused similar effects (Figure 2A,B).

For the Crumbs complex, some constituents regulate the nuclear enrichment of YAP and TAZ upon silencing (angiomotin-like 2 (AMOTL2), Lin-7 homolog C, Crumbs cell polarity complex component (LIN7C), MPP5) (Figure 2B). Notably, inhibiting other apical polarity proteins was associated with the strong nuclear enrichment of only one Hippo pathway effector (e.g., Crumbs cell polarity complex component 3 (CRB3) deficiency caused the nuclear shuttling of TAZ but not YAP) (Figure 2A). This inconsistent effect of Crumbs constituents on YAP or TAZ might be due to the strict analysis of nuclear YAP/TAZ accumulation (at least 1/3 of all analyzed cells must exhibit nuclear YAP or TAZ positivity). Alternatively, the different protein structures of YAP and TAZ may influence their binding properties to Crumbs proteins. For instance, TAZ lacks one WW domain with a PPxY binding motif [20].

In summary, inhibiting apical Crumbs complex proteins consistently causes the nuclear enrichment of YAP and/or TAZ in liver cancer cells. Other cell polarity complex constituents are less critical under the chosen experimental conditions.

### 2.2. Confirming the Interaction of YAP and TAZ with MPP5

MPP5 caught our attention as the inhibition of this factor robustly increases nuclear YAP/TAZ in our model system (Figure 2B). To confirm the physical binding of endogenous YAP and TAZ with MPP5, we performed co-immunoprecipitation (Co-IP) experiments. Notably, clear interaction with MPP5 was detected only for YAP after precipitating endogenous YAP or TAZ (Figure 3A). Because low protein levels may account for insufficient interaction between these proteins, we overexpressed YAP or TAZ in HepG2 cells. In this case, an apparent binding of MPP5 to YAP and TAZ was detectable (Figure 3B).

We then asked which MPP5 protein domains might be relevant to the binding of Hippo pathway effectors. To answer this question, we focused on YAP because a robust interaction between endogenous MPP5 and YAP was observed (Figure 3A). We created MPP5 mutants lacking domains that are important for protein–protein interactions: MPP5ΔL27 (lacking the L27 domain), MPP5ΔPDZ (lacking the PSD-95-Dlg1-ZO-1 (PDZ) domain), MPP5ΔSH3 (lacking the Src-Homology 3 (SH3) domain), and MPP5ΔGuKc (lacking the Guanylate Kinase, catalytic domain (GuKc) domain) (Figure 3C) [21,22]. Transient transfection of the vectors illustrated the expression of all isoforms, with MPP5ΔPDZ showing one instead of two expected MPP5 protein isoforms (Figure 3D). Subsequent co-expression of HA-tagged YAP and the MPP5 protein mutants followed by precipitation of YAP and detection of Flag-tagged MPP isoforms demonstrated that the MPP5ΔPDZ loses the ability to bind YAP in vitro (Figure 3E). In addition, a reduction in interaction with YAP was also observed, particularly for MPP5ΔSH3 and MPP5ΔGuKc, suggesting that the loss of these domains also affects the structure of MPP5 and its ability for protein interaction.

In summary, YAP and TAZ physically interact with MPP5 predominantly via the PDZ domain.

### 2.3. The Inactivation of MPP5 Enables the Nuclear Transport and Activation of YAP

MPP5 is part of the apical cell polarity complex and should localize close to the cell membrane; however, IF analyses illustrated a moderate to strong cytoplasmic enrichment of endogenous MPP5 in different liver cancer cell lines (HepG2, HuH6, HLF) (Figure 4A). In addition, an intense MPP5 membrane staining was observed in HuH6 cells and a weak staining in HepG2 cells. This indicates that partial or complete MPP5 mislocalization is frequent in HCC cells, which does not exist in the highly structured liver environment characterized by apical, basolateral, and lateral hepatocellular orientation. To demonstrate if MPP5 interacts with YAP at the membrane or in the cytoplasm of these cell lines, a proximity ligation assay (PLA) was performed. The PLA results showed a predominant cytoplasmic MPP5/YAP interaction, although a moderate membranous co-localization of both proteins was also evident in HuH6 cells (Figure 4B).

Next, we aimed to verify our functional screen data indicating that the inactivation of cytoplasmic MPP5 may facilitate the nuclear translocation and activation of YAP (Figure 2A,B). Indeed, the efficient inhibition of MPP5 by two siRNAs led to a moderate reduction in YAP phosphorylation, which serves as a proxy for its activation (phosphorylated YAP is predominantly detectable in the cytoplasm while dephosphorylation leads to its nuclear transport) (Figure 4C) [23]. Protein fractionation experiments confirmed the regulatory impact on YAP after MPP5 inhibition. Here, the reduction in MPP5 expression led to a detectable accumulation of YAP in the nuclear protein fraction of liver cancer cells (Figure 4D).

Finally, we asked whether the nuclear YAP enrichment, caused by MPP5 inhibition, could induce a transcriptional response specific to YAP. For this, we inhibited MPP5 using siRNAs and performed expression profiling analysis. Subsequently, we checked the expression of two gene signatures known to be transcriptionally regulated by YAP (called Wang and Cordenonsi signatures) [24,25]. Indeed, the induction of many YAP target genes from both signatures was observed following MPP5 silencing, further supporting a nuclear accumulation and activation of YAP (Figure 4E).

To summarize, MPP5 engages with YAP, inhibiting its movement to the nucleus and activation in liver cancer cells.

### 2.4. The Loss of MPP5 in Human HCCs Is Associated with the Activation of YAP and Poor Clinical Outcome

To analyze the connection between MPP5 and the Hippo signaling pathway in liver cancer patient samples, tissue-micro arrays (TMAs) containing human HCCs and healthy liver tissues were stained for MPP5, YAP, TAZ, the YAP/TAZ target gene mini-chromosome maintenance protein 2 (MCM2), and the proliferation marker Ki-67 [26].

In healthy liver tissue, MPP5 showed a distinct membranous stain at the hepatocyte apical (canalicular) membrane (Figure 5A). This membranous localization was detected to a lesser extent in well-differentiated (G1 and G2) tumors and gradually disappeared with tumor dedifferentiation (G3 and G4). As known from the literature, high-grade tumors displayed a significant increase in YAP/TAZ positivity in HCCs compared to normal liver tissue and well-differentiated HCCs (Figure 5A,B) [16,26]. Equally, MCM2 and Ki-67 showed a low nuclear expression in normal liver tissue and low-grade tumors and exhibited a robust nuclear stain in high-grade tumors. Statistical analysis revealed that membranous MPP5 was negatively associated with tumor dedifferentiation (r_s_ = −0.401, *p* ≤ 0.001). In contrast, nuclear YAP (r_s_ = 0.263, *p* ≤ 0.01) and TAZ (r_s_ = 0.527, *p* ≤ 0.001), MCM2 (r_s_ = 0.444, *p* ≤ 0.001), and Ki-67 stains (r_s_ = 0.427, *p* ≤ 0.001) positively correlated with tumor grading. Notably, membranous MPP5 had a moderate negative correlation with nuclear YAP (r_s_ = −0.214, *p* ≤ 0.04) and TAZ (r_s_ = −0.480, *p* ≤ 0.001), MCM2 (r_s_ = −0.326, *p* ≤ 0.001), and Ki-67 (r_s_ = −0.202, *p* ≤ 0.05).

Next, we analyzed published transcriptome patient data from non-malignant liver tissues (*n* = 168) and HCCs (*n* = 228) [27]. Interestingly, *MPP5* transcript levels were not reduced in HCCs but showed increased expression compared to non-malignant livers (Figure 6A). However, due to a significant variation in *MPP5* expression within the HCCs, a reduction in *MPP5* expression can be detected in about 16.2% of all patients (compared to the non-malignant liver tissues (Figure 6B). This indicated that the observed loss of membranous MPP5 in HCC tissues is not exclusively mediated by transcriptional regulation but also by post-translational effects or subcellular mislocalization. As indicated by our in vitro experiments after MPP5 inhibition, a moderate negative statistical correlation between *MPP5* expression and the induction of YAP/TAZ target genes was evident in the group of HCCs (Figure 6C) [24,25]. A comparable negative association was observed in an independent HCC cohort (Appendix A) [28]. Notably, the low expression of *MPP5* in HCCs statistically correlated with poor overall and disease-free survival (Figure 6D). No statistical relationship could be found between *MPP5* expression and other clinical parameters of the cohort.

Together, these findings illustrate that MPP5 is losing its proper apical localization in hepatocarcinogenesis, which correlates with nuclear YAP/TAZ. Moreover, those HCC patients with reduced MPP5 levels are characterized by poor clinical outcomes and elevated YAP/TAZ activity.

## 3. Discussion

The Hippo pathway plays a pivotal role in organ development, regeneration, and, in cases of dysregulation, cancer development. Epithelial cell polarity and other upstream regulators control Hippo pathway activity, such as G-protein-coupled receptor signaling, cell–cell adhesion, and mechanotransduction. Moreover, the aberrant nuclear accumulation of YAP and TAZ is common in various cancers, such as HCC [24,26]. In this study, we employed a systematic approach to identify polarity complexes and their constituents that affect the localization and activity of both Hippo pathway effectors YAP and TAZ in liver cancer cells.

The Crumbs complex is an apical polarity determinant defining the spatial organization of highly polarized cells. In addition, its constituents contribute to regulating cell signaling pathways, as illustrated by the phosphoinositide-3-kinase (PI3K)/AKT serine/threonine kinase (AKT) pathway [29]. Notably, our data demonstrate that several Crumbs complex proteins may directly or indirectly interact with YAP and TAZ in HCC cells. Therefore, the Crumbs complex functions as a structural hub that links and coordinates spatial cell organization with the activity of various signaling pathways [30,31]. Earlier studies confirmed our findings, indicating that components of this complex, like CRB3, directly engage with YAP/TAZ or regulate the activity of upstream Hippo pathway factors, including LATS1/2 [32,33]. Independent of the mechanism, these interactions at the membrane or in the cytoplasm lead to the nuclear exclusion of YAP and TAZ. This highlights the importance of cellular polarity as a tumor-suppressive characteristic of cells [34,35].

Here, we show that the Crumbs complex component MPP5 inhibits YAP and TAZ by sequestering them away from the nucleus. Predominantly via the PDZ domain, MPP5 functions as a spatial regulator that can control YAP/TAZ localization and activity without significantly affecting their total levels. This is confirmed by findings illustrating that the PDZ-binding motif of MPP5 and the SH3 domain of YAP/TAZ are essential for interacting with the Crumbs complex proteins, including MPP5 [31]. Notably, our results further suggest that MPP5 retains its YAP/TAZ binding capacity even in cells characterized by a cytoplasmic enrichment of MPP5. This mislocalization of MPP5 is most likely due to the absence of tissue context in carcinogenesis, which influences cell polarity and cell–cell junction formation. However, our data also suggest that the loss of MPP5 alone leads to only a limited induction of YAP and TAZ in the tissue context. While MPP5 inhibition in vitro effectively results in the nuclear localization of YAP and TAZ, the induction of YAP/TAZ target genes in the tissue context is moderate. It is conceivable that the loss of MPP5 in tissues elicits a limited response because other Crumbs polarity factors may compensate for YAP/TAZ retention.

For various tumor entities, such as colorectal cancer, it has been reported that the reduction in MPP5 mediates pro-tumorigenic properties and is correlated with poor survival in cancer patients [36,37]. In contrast, increased *MPP5* mRNA levels are associated with shortened metastasis-free survival for metastatic breast cancer [38]. These partly contradicting observations suggest that MPP5 dysregulation may affect various tumor entities differently. Interestingly, we also observed an increase, rather than a reduction, of *MPP5* expression at the mRNA level in HCC patients. However, the situation appears different at the protein level, as we detected a loss of membrane expression at cell contact sites in liver tissue. Simultaneously, many liver cancer cell lines exhibit prominent cytoplasmic expression of MPP5, which, despite its incorrect subcellular localization, still retains the ability to bind YAP. In this context, the origins of cells could also affect the expression and subcellular localization of MPP5. For instance, both HuH6 and HepG2 cells were derived from hepatoblastomas, whereas HLF cells represent HCC cells. Since hepatoblastoma is a tumor that originates from immature precursor cells of the liver, whereas HCC cells dedifferentiate from hepatocytes, the differences in MPP5 expression and localization may be attributed to the origins of the cell lines examined in our study. Thus, various molecular mechanisms, such as aberrant localization and aberrant expression, appear to contribute to the tumor-relevant properties of MPP5. The ability of MPP5 to bind and sequester YAP/TAZ suggests tumor-suppressive properties in liver cancer. This is supported by the analysis of patient samples, where low expression of *MPP5* in HCCs is significantly associated with poor survival. At the same time, low *MPP5* expression correlates with YAP/TAZ-induced gene signatures [24,25] and poor prognosis for cancer patients.

If the loss of cellular polarity promotes tumorigenesis, therapies that restore spatial polarity could be effective. Indeed, various approaches aim to reconstruct cellular polarity or target downstream signaling pathways. For example, restoring murine *Crb3* improved cell polarity and reduced tumor cell properties in kidney epithelial cells [39]. Alternatively, the activity of aberrantly active signaling pathways can be pharmacologically inhibited. This has been successfully demonstrated for the PI3K/AKT and Hippo/YAP/TAZ pathways. For example, the FDA and EMA-approved Alpelisib specifically blocks PI3Kα, while recent research focuses on disrupting the YAP/TAZ interaction with TEAD transcription factors [9,10,40]. To our knowledge, no substances have yet been identified to restore compromised polarity in tumor cells. Consequently, therapeutic options for reestablishing cellular polarity appear limited to genetic engineering approaches or blocking downstream mechanisms.

In summary, spatial alterations in cell polarity, commonly observed in tumor cells, significantly impact the activity of tumor-relevant signaling pathways. We demonstrate that the Crumbs complex component MPP5 is an important inhibitor of YAP/TAZ activity in hepatocarcinogenesis.

## 4. Materials and Methods

The siRNA sequences, primers, and antibodies used in this study can be found in the Appendix A.

### 4.1. Cultivation of Cells, siRNA and Vector Transfections

The media for the liver cancer cell lines HepG2 (RPMI-1640 medium), HuH6 (DMEM medium), and HLF (DMEM medium) were supplemented with 10% FCS (Thermo Fisher, Darmstadt, Germany) and 1% penicillin/streptomycin (Sigma-Aldrich, Steinheim, Germany). HepG2 and HuH6 cells are cell lines derived from hepatoblastoma, whereas HLF cells originated from HCC. All investigated cells were positive for YAP and TAZ, as demonstrated previously [16].

Cells were maintained at 37 °C in a humidified atmosphere containing 5% CO_2_. They were regularly checked for mycoplasma contamination and authenticated by short tandem repeat (STR) analysis (Microsynth, Göttingen, Germany).

SiRNAs were transfected using the Lipofectamine RNAiMAX transfection reagent (Thermo Fisher) according to the manufacturer’s instructions. Protein and total RNA were harvested after 48 h.

A customized siRNA FlexiPlate library targeting transcripts of 26 different proteins involved in hepatocellular polarity was ordered from Qiagen (Qiagen, Hilden, Germany). At least two siRNAs were tested for each gene (final concentration: 30 nM, sequences listed in Appendix A). Unspecific siRNA was used as a negative control, while LATS1-specific siRNAs were used as positive controls. *MPP5*, *INADL*, and *LIC7C* expression were tested to confirm transfection efficiency (Appendix A). Real-time PCR was previously described [15].

For vector transfection, cells were seeded in a 10 cm dish and cultured until they reached 80% confluence. FUGENE^®^ was used as a transfection reagent according to the manufacturer’s protocol (Promega, Walldorf, Germany).

### 4.2. Gateway Cloning

Cloning of *YAP* and *TAZ* cDNA into the lentiviral pTRIPz vector has been described previously [15]. Primers were designed to delete MPP5 domains, flanking the domain to be deleted. After product amplification using the Q5 high-fidelity DNA polymerase (New England Biolabs, Frankfurt a. Main, Germany), the original template was digested with DpnI. The newly synthesized *MPP5* cDNA was used for cloning into the pDEST-EGFP-N vector. All constructs were sequenced to verify accuracy.

### 4.3. BioID Assay and Lc-MS Analysis

The execution of the BioID assay, the subsequent LC-MS analysis, and the bioinformatic analysis were previously described [15]. The data have been deposited in the Proteomics Identification Database (PRIDE) (https://www.ebi.ac.uk/pride/; ID: PXD058235; accessed on 26 November 2024).

### 4.4. Protein Isolation, Nuclear/Cytoplasmic Fractionation, and Western Immunoblotting

Cell harvesting and Western Immunoblotting have recently been described [41]. The NE-PER^TM^ Nuclear and Cytoplasmic Extraction Kit was utilized according to the manufacturer’s instructions to isolate subcellular protein fractionations. Western blot detection and quantification were performed using the Odyssey-CLx Infrared Imaging system with the ImageStudio software V5.2 (LI-COR Biosciences, Bad Homburg, Germany). Glyceraldehyde-3-phosphate dehydrogenase (GAPDH) or β-Actin were used as loading controls for total protein fractions, while PARP and β-tubulin served as fractionation controls. Co-IP experiments were performed as previously described [26].

### 4.5. Proximity Ligation Assay (PLA)

Coverslips were placed in 24-well plates and sterilized under UV light for 20 min. PLA was performed using the Duolink^®^ Proximity Ligation Assay Kit (Sigma-Aldrich). Cells seeded on coverslips were washed with phosphate-buffered saline (PBS) containing 2 mM MgCl_2_ after 24 h. Cells were subsequently fixed with 4% paraformaldehyde for 10 min and washed four times with PBS for 5 min, followed by permeabilization for 5 min with 0.1% Triton in PBS. After washing twice with PBS for 5 min, coverslips were transferred into a wet chamber. Blocking, primary antibody incubation, probe incubation, ligation, and amplification were performed according to the manufacturer’s instructions. After the final washing steps, the coverslips were mounted onto a microscopy slide using 4′,6-diamidin-2-phenylindol (DAPI) Fluoromount-G (Southern Biotech, Birmingham, UK). Fluorescence microscopy was carried out using the Keyence microscope (BZ-X, Keyence GmbH, Neu-Isenburg, Germany). Single antibody incubations were used as controls.

### 4.6. Immunofluorescence Analysis of Cultured Cells

Coverslips were placed in 24-well plates and sterilized under UV light for 20 min. For the immunofluorescence staining, the medium was removed, and the cells were washed with 200 µL PBS containing 2 mM MgCl_2_. After discarding the PBS solution, the cells were fixed in paraformaldehyde for 15 min. The fixation was followed by three wash steps with 600 µL PBS. The permeabilization was performed with 0.2% Triton/PBS solution at room temperature for 7 min. After permeabilization, the cells were washed three times (3 × 600 µL PBS). The coverslips were blocked with 600 µL 0.5% bovine serum albumin (BSA) solution at room temperature for 30 min to prevent unspecific staining. The coverslips were then placed on Parafilm and 100 µL PBS containing the first antibody was added to each coverslip. After incubation in a humid chamber at room temperature for 1.5 h, the coverslips were washed 3 × 5 min in 0.01% TWEEN/PBS solution. Afterward, the coverslips were incubated with the secondary antibodies in a humid chamber at room temperature for 1 h. The coverslips were washed 3 × 5 min in 0.01% TWEEN/PBS solution, rinsed with Millipore, and transferred in 100% ethanol for 3 min. The coverslips were dried at room temperature and covered with 20–30 µL Fluoromont/DAPI. Fluorescence microscopy was carried out using the Olympus IX81 microscope and the Hamamatsu ORCA-R2 camera (Olympus, Hamburg, Germany).

### 4.7. Tissue Processing, Tissue-Microarray (TMA), and Immunohistochemistry

Immunohistochemical stains for YAP, TAZ, MCM2, and Ki-67 have been described previously [16,26].

The human HCC TMA used for immunohistochemical analysis of MPP5, YAP, TAZ, MCM2, and Ki-67 contained unrelated normal liver tissues (*n* = 7) and HCCs (*n* = 105) with well to poorly differentiated tumors (grading: G1 = 10, G2 = 75, G3 = 16, G4 = 4). For each tissue sample and staining, an evaluation score was derived based on the following scoring system: quantity (0 = no expression; 1 = less than 1% positive cells; 2 = 1–9% positive cells; 3 = 10–50% and 4 = more than 50% positive cells) and intensity (0 = negative; 1 = low; 2 = medium; 3 = strong). The final score was calculated by multiplying quantity and intensity (range: 0–12). To quantify in situ hybridization of the lncRNA Morrbid, stained mouse liver tissue sections were digitalized using digital slide scanners (Aperio AT2, Leica Mikrosysteme Vertrieb GmbH, Wetzlar, Germany) at 40× magnification.

### 4.8. Patient Expression Data and Data Analysis

Expression data for human non-malignant livers and HCCs was derived from two cohorts. Cohort 1 consists of 168 liver tissues and 228 HCCs (GSE63898) [27], and cohort 2 contains 239 liver tissues and 247 HCCs (GSE14520) [28].

Two target gene lists were used to calculate signature scores that serve as a proxy for YAP/TAZ activity [24,25]. The Wang signature score consists of 22 genes (2 genes were missing in cohort 2) and the Cordenonsi signature consists of 57 genes. Subsequently, an equally weighted score for all genes in both signatures was calculated for each patient. This score was used for statistical correlation with *MPP5* expression in HCC samples.

For overall survival and cancer recurrence analysis, HCC patients from cohort 2 were stratified into two groups (high and low) using the median as the cutoff. Differences in patient survival were assessed using the log-rank test and presented as Kaplan–Meier curves.

### 4.9. Expression Profiling

To identify MPP5-regulated genes, HepG2 cells were transfected with two *MPP5*-specific siRNAs (30 nM). Total RNA was isolated 24 h after transfection using the NucleoSpin RNA II kit (Macherey-Nagel, Düren, Germany). Only samples with an RNA integrity number (RIN) > 7 were considered for microarray analysis.

Gene expression profiling was performed using human HuGene-2_0-st arrays (Thermo Fisher Scientific). Biotinylated antisense cDNA was prepared according to the standard labeling protocol with the GeneChip^®^ WT Plus Reagent Kit and the GeneChip^®^ Hybridization, Wash and Stain Kit (Thermo Fisher Scientific). Afterward, the hybridization on the chip was performed on a GeneChip Hybridization oven 640, dyed in the GeneChip Fluidics Station 450, and scanned with a GeneChip Scanner 3000. The equipment was provided by Affymetrix (Affymetrix, High Wycombe, UK).

A Custom CDF Version 21 with ENTREZ-based gene definitions was used to annotate the arrays. The raw fluorescence intensity values were normalized by applying quantile normalization and RMA background correction. OneWay-ANOVA was performed to identify differentially expressed genes using a commercial software package SAS JMP15 Genomics, version 10 (SAS Institute, Cary, NC, USA). An FDR of a = 0.05 with correction was considered the significance level. The raw and normalized data are deposited in the Gene Expression Omnibus database (http://www.ncbi.nlm.nih.gov/geo/; accession no. GSE282884; accessed on 26 November 2024).

### 4.10. Statistics and Software

Data are represented as mean −/+ standard deviation. The non-parametric Mann–Whitney U test was used for the statistical comparison of the two groups, while Spearman’s rank correlation coefficient (r_s_) was used for correlation analysis. All statistical tests were performed with GraphPad Prism 10 software. Significance levels are as follows: * *p* ≤ 0.05, ** *p* ≤ 0.01, *** *p* ≤ 0.001. Quantification of Western blot signals was carried out with Fiji (ImageJ2 Version 2.14.0) [42].

## Figures and Tables

**Figure 1 ijms-26-00660-f001:**
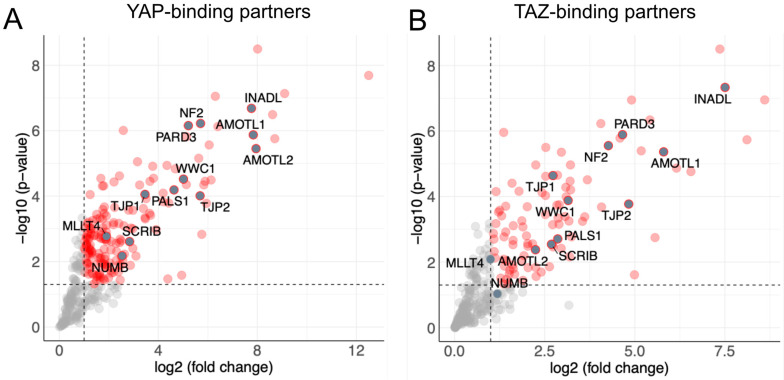
Identification of cell polarity proteins interacting with YAP and TAZ in liver cancer cells. Volcano plot showing a significant binding of proteins to YAP (**A**) or TAZ (**B**) identified by LC-MS analysis. Proteins known to contribute to cell polarity are indicated. The vertical line indicates a two-fold enrichment for binding interactions. The horizontal line represents a false discovery rate (FDR) of *p* ≤ 0.05.

**Figure 2 ijms-26-00660-f002:**
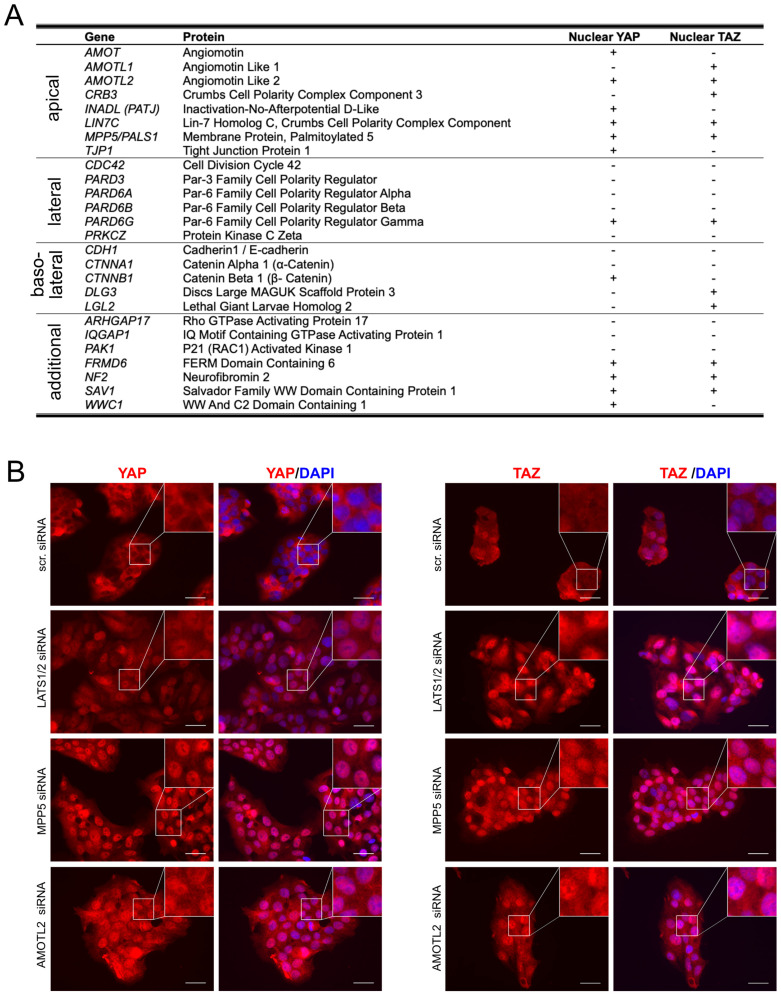
Functional screen for cell polarity factors that regulate YAP and TAZ. (**A**) Table summarizing proteins contributing to cell polarity (e.g., apical, lateral, and basolateral complex constituents). Two siRNAs for each gene were transfected in HepG2 cells, and endogenous YAP or TAZ localization was investigated utilizing immunofluorescence. (+) = more than 1/3 of cells show a strong nuclear YAP/TAZ enrichment with both siRNAs; (-) = less than 1/3 of cells or weak nuclear enrichment was observed. (**B**) Exemplary pictures of immunofluorescent YAP and TAZ stains are shown after the transfection of scrambled siRNA (scr. siRNA, negative control), Large tumor suppressor kinase 1/2 siRNAs (positive control), MPP5 siRNAs and AMOTL2 siRNAs for 48 h. Bars: 25 µm.

**Figure 3 ijms-26-00660-f003:**
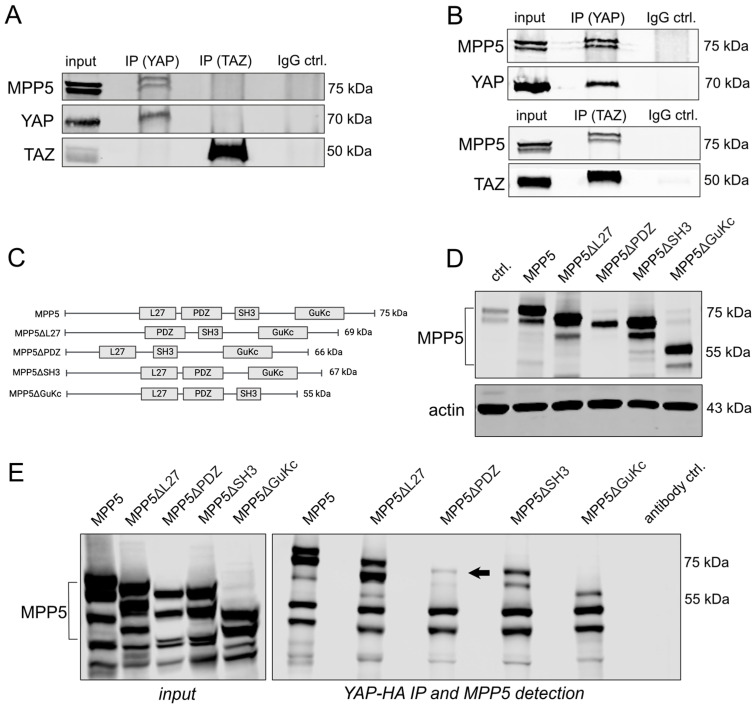
MPP5 physically interacts with YAP/TAZ. (**A**) Co-IP with endogenous YAP, TAZ, and MPP5 proteins in HepG2 cells. (**B**) Co-IP after overexpression of YAP or TAZ (upper and lower blot, respectively) for 48 h in HepG2 cells. (**C**) Scheme of human MPP5 and mutant isoforms used for Co-IP experiments. (**D**) Western immunoblot analysis confirmed the efficient expression of all MPP5 isoforms. The control shows the endogenous MPP5 expression (ctrl.). (**E**) Co-IP experiment after overexpression of HA-tagged YAP and all MPP5 isoforms for 48 h. MPP5 is detected after the pull-down of YAP using an HA-specific antibody. The arrow indicates the position where the MPP5 isoform was expected. Total protein lysate was used as input control for (**A**,**B**,**E**). Rabbit serum IgG served as negative control (IgG ctrl./antibody ctrl.).

**Figure 4 ijms-26-00660-f004:**
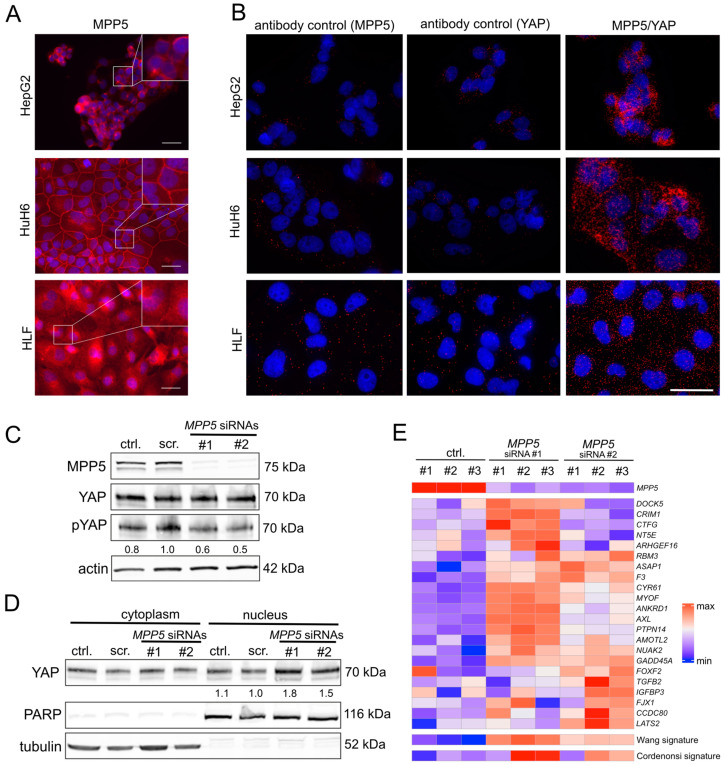
MPP5 is a negative regulator of YAP activity. (**A**) IF microscopy detecting endogenous MPP5 in HepG2, HuH6, and HLF cells. The cell density was chosen to create both dense and subconfluent areas. Scale bars: 25 µm. (**B**) PLA for YAP and MPP5 in different liver cancer cell lines. MPP5 and YAP alone represent negative controls, while the combined administration illustrates their cytoplasmic and partly membranous co-localization. (**C**) Western immunoblot of total protein fractions after MPP5 inhibition using two siRNAs (#1 and #2). (**D**) Western immunoblot of cytoplasmic and nuclear protein fractions after MPP5 inhibition. Poly (ADP-ribose) polymerase (PARP) and tubulin were fractionated controls. (**E**) Heatmap illustrating the expression of YAP target genes (Wang signature, *n* = 22) after inhibition of MPP5. The efficient knockdown of *MPP5* is illustrated in the first line. The lower section of the heatmap displays a balanced score for two YAP-dependent gene signatures (Wang signature and Cordenonsi signature, *n* = 57). For (**C**–**E**), untreated cells (ctrl.) and scrambled siRNA (scr.)-transfected cells were used as controls. For Western blot quantification, pYAP and YAP signals were measured with the Fiji software (ImageJ2 Version 2.14.0) and normalized to endogenous YAP (**C**) or nuclear PARP (**D**). The results of signal quantification are shown under the respective protein panels.

**Figure 5 ijms-26-00660-f005:**
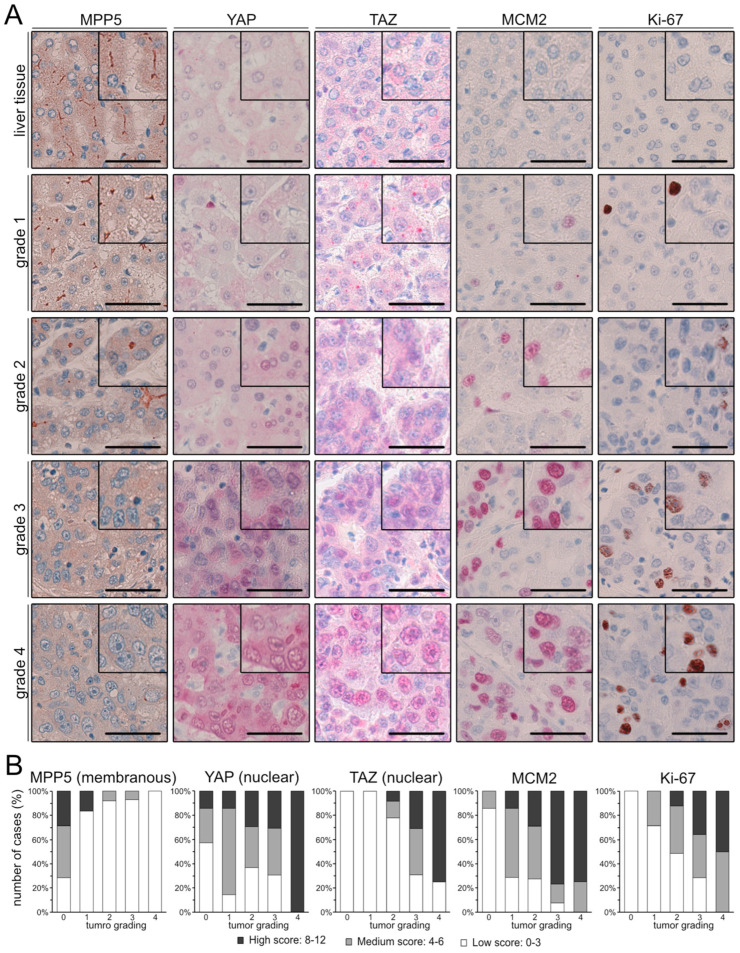
Membranous MPP5 expression negatively correlates with nuclear YAP/TAZ enrichment in HCC cells. (**A**) Immunohistochemical analysis of MPP5, YAP, TAZ, MCM2, and Ki-67. Higher magnifications are shown in the upper right corner. Scale bars: 50 µm. (**B**) Proportional bar charts illustrate the correlation of IHC stains with tumor grading. IHC scores (white: low score, grey: medium score, black: high score) were correlated with tumor grading (0: normal liver tissue, 1: G1, 2: G2, 3: G3, 4: G4). Spearman’s correlation was used for statistical testing.

**Figure 6 ijms-26-00660-f006:**
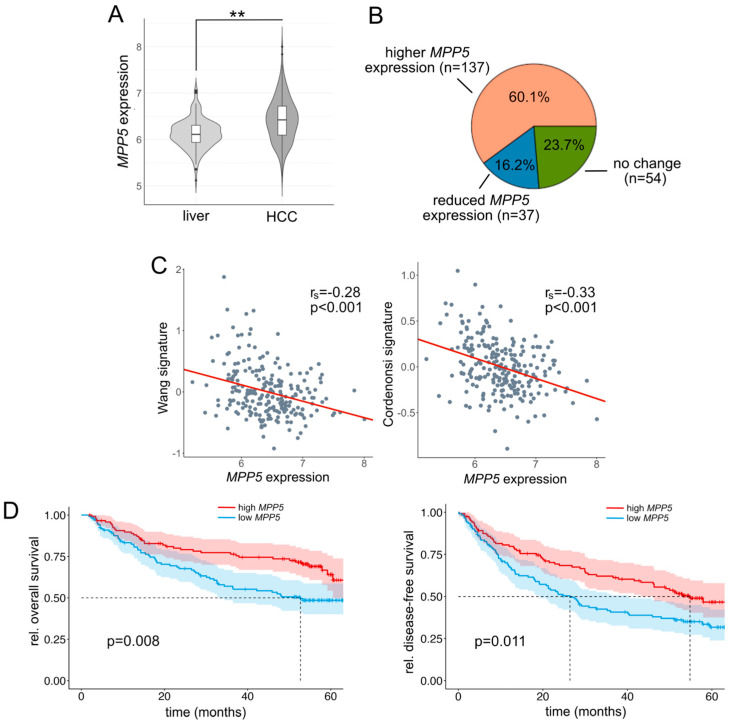
*MPP5* negatively correlates with YAP target gene expression and clinical outcome in HCC patients. (**A**) Transcriptome analysis of *MPP5* mRNA levels in HCC tissues compared to surrounding liver tissues. Mann–Whitney U test (** *p* ≤ 0.01). (**B**) Distribution of MPP5 expression in HCCs. Tumor tissues with MPP5 levels lower than 75% of non-malignant tissues were classified as ‘reduced’. (**C**) Association between MPP5 and YAP target gene signature expression (Wang and Cordenonsi signatures). Spearman correlation analysis. (**D**) Higher MPP5 transcript levels (red) significantly correlate with better overall survival and disease-free survival of HCC patients. Respective Kaplan–Meier curves are shown.

## Data Availability

The paper and the Appendix A contain all necessary information to assess the conclusions. The corresponding author will fulfill requests for resources and reagents. The proteomics data have been deposited in the Proteomics Identification Database (PRIDE) (https://www.ebi.ac.uk/pride/; ID: PXD058235; accessed on 26 November 2024). Raw and normalized data were deposited in the Gene Expression Omnibus database (https://www.ncbi.nlm.nih.gov/geo/; GSE282884; accessed on 26 November 2024).

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
