# Peer review of "The Cell Polarity Protein MPP5/PALS1 Controls the Subcellular Localization of the Oncogenes YAP and TAZ in Liver Cancer"

_ijms, 2025, doi:10.3390/ijms26020660_

Round 1

Reviewer 1 Report

Comments and Suggestions for Authors

Comments to the Authors

The manuscript from Marcell Tóth et al. entitled “The cell polarity protein MPP5/PALS1 controls the subcellular localization of the oncogenes YAP and TAZ in hepatocellular carcinoma” (Manuscript ID: ijms-3385530) seems interesting. There are points which need to be addressed.

1.       The cell line HepG2 seems under debate as to whether it is hepatocellular carcinoma or not. Why did the authors choose the cell? (doi: 10.1016/j.humpath.2009.07.003.) Also, what about other cell lines used in this study. Are those appropriate as HCC?

2.       Each of the three cells, HepG2, Huh6, and HLF, has their own characteristics, including their origin. Any differences with respect to the localization and activity of YAP/TAZ? Authors can discuss.

Author Response

Feedback: The manuscript from Marcell Tóth et al. entitled “The cell polarity protein MPP5/PALS1 controls the subcellular localization of the oncogenes YAP and TAZ in hepatocellular carcinoma” (Manuscript ID: ijms-3385530) seems interesting. There are points which need to be addressed.

Reply: We thank the reviewer for his/her positive feedback. We are happy to address the comments.

Comment #1: The cell line HepG2 seems under debate as to whether it is hepatocellular carcinoma or not. Why did the authors choose the cell? (doi: 10.1016/j.humpath.2009.07.003.; PMID 19751877). Also, what about other cell lines used in this study. Are those appropriate as HCC?

Reply comment #1: The reviewer is correct in stating that HepG2 cells are not HCC cells but hepatoblastoma cells, as indicated in the reviewer's reference [1]. The HuH6 cell line also originates from hepatoblastoma cells, whereas the HLF cell line was established from an HCC [2, 3]. In fact, throughout the manuscript, we consistently refer to these cell lines as 'liver cancer cells' to avoid any misleading terminology (lines 190 in Results and 361 in Materials & Methods).

We selected HepG2 cells for most investigations because they exhibit many hepatocyte-specific functions (see lines 118–120). Furthermore, they display a notably high cell polarity [4, 5]. Since our studies on MPP5 suggested an impact on cell polarity, we have chosen HepG2 cells. In this context, HuH6 and HLF cells were utilized as alternative in vitro model systems, showcasing varying degrees of polarity and different MPP5 localization.

To avoid any misunderstanding regarding the cell lines, we adjusted sections of our revised manuscript and added a new reference (line 119, lines 363-366).

Comment #2: Each of the three cells, HepG2, Huh6, and HLF, has their own characteristics, including their origin. Any differences with respect to the localization and activity of YAP/TAZ? Authors can discuss.

 Reply comment #2: In a previous study, we comparatively analyzed the expression of YAP and TAZ in the cell lines HepG2, HLF, and HuH6, among others [6]. All cell lines express significant levels of YAP and TAZ and respond to low cell density with nuclear enrichment of both factors. The functionality of the Hippo signaling pathway in these cell lines was also a reason for selecting them for this study.

In response to the reviewer’s query, we have revised the manuscript accordingly and included the relevant reference.

References

  1. Lopez-Terrada, D.; Cheung, S. W.; Finegold, M. J.; Knowles, B. B., Hep G2 is a hepatoblastoma-derived cell line. Hum Pathol 2009, 40, (10), 1512-5.
  2. Doi, I., Establishment of a cell line and its clonal sublines from a patient with hepatoblastoma. Gan 1976, 67, (1), 1-10.
  3. Dor, I.; Namba, M.; Sato, J., Establishment and some biological characteristics of human hepatoma cell lines. Gan 1975, 66, (4), 385-92.
  4. Arzumanian, V. A.; Kiseleva, O. I.; Poverennaya, E. V., The Curious Case of the HepG2 Cell Line: 40 Years of Expertise. Int J Mol Sci 2021, 22, (23).
  5. van IJzendoorn, S. C.; Zegers, M. M.; Kok, J. W.; Hoekstra, D., Segregation of glucosylceramide and sphingomyelin occurs in the apical to basolateral transcytotic route in HepG2 cells. J Cell Biol 1997,137, (2), 347-57.
  6. Weiler, S. M. E.; Lutz, T.; Bissinger, M.; Sticht, C.; Knaub, M.; Gretz, N.; Schirmacher, P.; Breuhahn, K., TAZ target gene ITGAV regulates invasion and feeds back positively on YAP and TAZ in liver cancer cells. Cancer Lett 2020, 473, 164-175.

Reviewer 2 Report

Comments and Suggestions for Authors

The authors are to be congratulated on their extensive work, however some issues require clarification.

Whether this is a literature review, an expert opinion, or a position paper, it should be better defined alongside a materials and methods section, which is lacking. 

The authors should define the experimental setting upon which they built their data. They alluded to a mouse model in their introduction, but this has to be clarified. 

In the introduction, the authors describe the role of YAP and TAZ based on data derived from mouse models. They should also put this in the context of human HCC or clarify whether experimental models can be transferred to the clinic.

The Results section introduces the use of HepG2 cells. I suggest the authors write a methods section outlining their experimental work. 

Is there clinical information on 228 patients with HCC? Were samples collected from tumors and livers or only from tumors? 

Is any information available on the outcome of HCC patients and if the YAP-TAZ pathway has a prognostic significance? 

Author Response

Feedback: The authors are to be congratulated on their extensive work, however some issues require clarification.

Reply: We thank the reviewer for his/her positive feedback. In the following, we are pleased to address all comments.

Comment #1: Whether this is a literature review, an expert opinion, or a position paper, it should be better defined alongside a materials and methods section, which is lacking. 

Reply comment #1: According to the journal's guidelines, this article shows experimental and previously unpublished data (original article). Regarding the Materials & Methods section, we would like to point out that it is located under Section 4, following the Discussion, according to the journal's requirements. In this section and the Supplementary Data, we have detailed all technical and experimental aspects of the manuscript to allow independent reproduction of the work.

Comment #2: The authors should define the experimental setting upon which they built their data.

Reply comment #2: As mentioned in the previous point, a comprehensive description of all experiments can be found in Section 4 of the manuscript. We hope this adequately addresses the reviewer's question.

Comment #3: They alluded to a mouse model in their introduction, but this has to be clarified. 

Reply comment #3: We understand the reviewer's point. We mention various mouse models in one sentence (lines 67-69), with the corresponding reference to these models provided in the following sentence. This may not be very clear.

We moved the respective references and added additional explanations (lines 67-71) to improve the clarity of these sentences.

Comment #4: In the introduction, the authors describe the role of YAP and TAZ based on data derived from mouse models. They should also put this in the context of human HCC or clarify whether experimental models can be transferred to the clinic.

Reply comment #4: Already in the introduction, before mentioning the respective mouse models, we highlighted the relevance of YAP and TAZ in liver cancer development (lines 63-65). To further emphasize the importance of the Hippo (YAP/TAZ) signaling pathway, as requested by the reviewer, we have added a sentence addressing its clinical significance and the development and testing of inhibitors (lines 65-67). In addition, a respective sentence was added in the first paragraph of the Discussion (lines 292-293).

Comment #5: The Results section introduces the use of HepG2 cells. I suggest the authors write a methods section outlining their experimental work. 

Reply comment #5: As noted in the replies to comments 1 and 2, all methods are detailed in Section 4 of the manuscript.

Comment #6: Is there clinical information on 228 patients with HCC?

Reply comment #6: The transcriptomic data mentioned by the reviewer have already been published, and the available clinical data, such as etiology, gender, and tumor size, have been specified in this previous manuscript [1]. However, our analyses did not reveal any statistical associations between MPP5 expression and these clinical parameters. In response to the reviewer’s comment, we have included this information in the revised version of the manuscript (lines 273-274).

Comment #7: Were samples collected from tumors and livers or only from tumors? 

Reply comment #7: For the above-mentioned transcriptomic analysis, both tumor and non-tumorous liver tissues from patients were used [1]. Independent liver and tumor samples were examined for immunohistochemical analysis. A clarifying word regarding this cohort has been added to the Methods section (line 440).

Comment #8: Is any information available on the outcome of HCC patients and if the YAP-TAZ pathway has a prognostic significance? 

Reply comment #8: In fact, there are several studies demonstrating the prognostic relevance of YAP and/or TAZ expression in HCC. We have referenced two of these studies in our manuscript [2, 3]. Due to space constraints, however, we refrained from including additional studies.

References

  1. Roessler, S.; Jia, H. L.; Budhu, A.; Forgues, M.; Ye, Q. H.; Lee, J. S.; Thorgeirsson, S. S.; Sun, Z.; Tang, Z. Y.; Qin, L. X.; Wang, X. W., A unique metastasis gene signature enables prediction of tumor relapse in early-stage hepatocellular carcinoma patients. Cancer Res 2010, 70, (24), 10202-12.
  2. Weiler, S. M. E.; Lutz, T.; Bissinger, M.; Sticht, C.; Knaub, M.; Gretz, N.; Schirmacher, P.; Breuhahn, K., TAZ target gene ITGAV regulates invasion and feeds back positively on YAP and TAZ in liver cancer cells. Cancer Lett 2020, 473, 164-175.
  3. Weiler, S. M. E.; Pinna, F.; Wolf, T.; Lutz, T.; Geldiyev, A.; Sticht, C.; Knaub, M.; Thomann, S.; Bissinger, M.; Wan, S.; Rossler, S.; Becker, D.; Gretz, N.; Lang, H.; Bergmann, F.; Ustiyan, V.; Kalin, T. V.; Singer, S.; Lee, J. S.; Marquardt, J. U.; Schirmacher, P.; Kalinichenko, V. V.; Breuhahn, K., Induction of Chromosome Instability by Activation of Yes-Associated Protein and Forkhead Box M1 in Liver Cancer. Gastroenterology 2017, 152, (8), 2037-2051 e22.

Round 2

Reviewer 1 Report

Comments and Suggestions for Authors

The re-submitted manuscript from Marcell Tóth et al. entitled “The cell polarity protein MPP5/PALS1 controls the subcellular localization of the oncogenes YAP and TAZ in hepatocellular carcinoma” seems interesting and revised appropriately somewhat. There are still but minor points which need to be addressed.

> Reply comment #1: The reviewer is correct in stating that HepG2 cells are not HCC cells but hepatoblastoma cells, as indicated in the reviewer's reference [1]. The HuH6 cell line also originates from hepatoblastoma cells, whereas the HLF cell line was established from an HCC [2, 3]. In fact, throughout the manuscript, we consistently refer to these cell lines as 'liver cancer cells' to avoid any misleading terminology.

Minor comments:

Given the author's argument, is it appropriate to use “in hepatocellular carcinoma” in the title? Please consider with the Editor whether to change them into "in liver cancer" or remove the words.

Author Response

Comment: Given the author's argument, is it appropriate to use “in hepatocellular carcinoma” in the title? Please consider with the Editor whether to change them into "in liver cancer" or remove the words.

Reply: We want to thank the reviewer for this comment. We agree and adjusted the manuscript's title according to his/her suggestion.